# Recent Advances in the Synthesis and Biological Activity of 8-Hydroxyquinolines

**DOI:** 10.3390/molecules25184321

**Published:** 2020-09-21

**Authors:** Haythem A. Saadeh, Kamal A. Sweidan, Mohammad S. Mubarak

**Affiliations:** 1Department of Chemistry, College of Science, United Arab Emirates University, Al Ain P.O. Box 15551, United Arab Emirates; h.saadeh@uaeu.ac.ae; 2Department of Chemistry, School of Science, The University of Jordan, Amman 11942, Jordan; k.sweidan@ju.edu.jo

**Keywords:** 8-hydroxyquinoline, synthesis, bioactivity, cancer, Alzheimer’s disease

## Abstract

Compounds containing the 8-hydroxyquinoline (8-HQ) **1** nucleus exhibit a wide range of biological activities, including antimicrobial, anticancer, and antifungal effects. The chemistry and biology of this group have attracted the attention of chemists, medicinal chemists, and professionals in health sciences. A number of prescribed drugs incorporate this group, and numerous 8-HQ- based molecules can be used to develop potent lead compounds with good efficacy and low toxicity. This review focusses on the recent advances in the synthesis of 8-HQ derivatives with different pharmacological properties, including anticancer, antiviral, and antibacterial activities. For this purpose, recent relevant references were searched in different known databases and search engines, such as MEDLINE (PubMed), Google Scholar, Science Direct, Scopus, Cochrane, Scientific Information Database (SID), SciFinder, and Institute for Scientific Information (ISI) Web of Knowledge. This review article provides a literature overview of the various synthetic strategies and biological activities of 8-HQ derivatives and covers the recent related literature. Taken together, compounds containing the 8-HQ moiety have huge therapeutic value and can act as potential building blocks for various pharmacologically active scaffolds. In addition, several described compounds in this review could act leads for the development of drugs against numerous diseases including cancer.

## 1. Introduction

8-Hydroxyquinoline derivatives are an important group of compounds with rich and diverse biological activities. These compounds incorporate the 8-hydroxyquinoline (8-HQ) moiety, which is a bicyclic compound that consists of a pyridine ring fused to phenol, in which the hydroxyl group is attached to position 8 [1]. In this respect, the pyridine ring maintains its properties as an electron-deficient entity with a basic nitrogen. In addition, and due to the presence of the phenolic group, 8-HQ displays typical phenolic properties that make it susceptible to numerous chemical reactions and structural modifications, such as electrophilic aromatic substitution, diazonium coupling, and molecular rearrangements. The close proximity of the hydroxyl group to the heterocyclic nitrogen makes 8-hydroxyquinolines good monoprotic bidentate chelating agents, which form four- and six-covalent complexes with a wide range of metal ions, including Cu^2+^, Zn^2+^, Bi^2+^, Mn^2+^, Mg^2+^, Cd^2+^, Ni^2+^, Fe^3+^, and Al^3+^ [2]. Figure 1 is a schematic representation of the different sites of reactions of 8-HQ. All of the aforementioned properties make 8-HQ a privileged structure with a variety of structural modifications, possessing a rich diversity of physical, chemical and biological properties.

8-Hydroxyquinoline has attracted the attention of chemists, medicinal chemists, and people in health sciences due to its unique physical and chemical properties. The interest in this compound and its derivatives has increased considerably in the last two decades [3]. 8-Hydroxyquinoline and many of its derivatives have a wide range of pharmacological applications, e.g., as iron-chelators for neuroprotection, as anticancer agents, as inhibitors of 2OG-dependent enzymes, as chelators of metalloproteins, as anti-HIV agents, as antifungal agents, as antileishmanial agents, as antischistosomal agents, as mycobacterium tuberculosis inhibitors, and as botulinum neurotoxin inhibitors [4]. Furthermore, these compounds are used as electron carriers in organic light-emitting diodes (OLEDs) and as fluorescent chemosensors for metal ions [5,6]. On the basis of the preceding discussion, and owing to the importance of 8-HQ from a chemistry point of view and to the wide range of biological activities and pharmacological applications of 8-HQ and derivatives, this review focuses on current knowledge of the synthetic methods of novel derivatives, along with the derivatives’ biological activities and medicinal applications. In addition, this review summarizes the most recent literature pertaining to the synthesis and bioactivity of 8-HQ and its derivatives, covering the period ranging from 2017 to the present. In addition, structure–activity relationships (SARs) of these derivatives are discussed to provide directions for further development of novel 8-HQ-based bioactive agents. For this purpose, recent relevant references have been obtained from different databases, such as Medical Literature Retrieval Analysis and Retrieval System Online (MEDLINE) (PubMed), Google Scholar, Science Direct, Scopus, Cochrane, SID, and SciFinder. Our intention is that the content and organization of this review will be valuable to the field and will greatly help researchers. Below are details about the recent documented synthesis methods of 8-HQ and its derivatives, as well as their biological activities against different diseases and disorders.

## 2. Antiviral Activity

Due to the Coronavirus 2019 (COVID-19) pandemic and its devastating economic, social, and health effects, we decided to start this study with the antiviral activities of some recent 8-HQ derivatives. A few publications have dealt with the antiviral activities of 8-HQ and its derivatives. De la Guardia et al. described the synthesis of novel 8-hydroxyquinoline derivatives (Figure 1) and investigated their activity against dengue virus [7]. These researchers prepared a number of compounds from 8-hydroxyquinoline N-oxide **2**, which upon treatment with copper-catalyzed Grignard reagents (RMgX; R = *i*-Pr and *i*-Bu) gave 2-alkyl-8-hydroxyquinoline **3**. Subsequent chlorination using N-chlorosuccinimide (NCS) under acidic conditions afforded the corresponding 2-alkyl-5,7-dichloro-8-hydroxyquinoline **4** in a good yield.

The antiviral activities of the two novel quinoline derivatives (R = *i*-Pr and *i*-Bu) **4** were evaluated in vitro against the dengue virus serotype 2 (DENV2). Both exhibited significant inhibitory activities against this virus. The results indicated that the *iso*-Pr-substituted derivative exhibits a half-maximal inhibitory concentration (IC_50_) of 3.03 µM and a half-maximal cytotoxic concentration (CC_50_) of 16.06 µM, for an estimated selectivity index (SI) of 5.30. On the other hand, the *iso*-Bu derivative was also active, showing a higher SI value of 39.5, with an IC_50_ of 0.49 µM and CC_50_ of 19.39 µM. The mechanism of action was also investigated and the results showed that these two derivatives are not virucidal, but appear to act at an early stage of the virus lifecycle, reducing the intracellular production of the envelope glycoprotein and the yield of infectious virions in treated and infected cells.

In a similar fashion, Kos and coworkers reported the microwave-assisted synthesis of thirty-two mono-, di-, and tri-substituted 8-hydroxyquinoline-2-carboxanilides, as shown in Figure 2 [8]. Condensation of activated 8-hydroxyquinoline-2-carboxylic acid (**5**) with substituted aniline **6** in the presence of by phosphorus trichloride yielded the desired target compound **7** in good amounts (61–79%).

These prepared compounds were subjected to bioactivity screening against the highly pathogenic H5N1 avian influenza viruses, with the results expressed as percentages of growth inhibition, and to cytotoxicity evaluation against the A549 cell line. The lipophilicity and electronic properties were the molecular parameters used to determine the structure–activity relationship. The lipophilicity of the studied compounds was determined as a log k value using reversed-phase high-performance liquid chromatography (RP-HPLC). Based on the presented results, most mono-substituted derivatives did not exert any antiviral activity or demonstrated only moderate activity. For example, 8-hydroxy-*N*-(3-nitrophenyl)quinoline-2-carboxamide showed optimum virus growth inhibition activity and cytotoxicity values of 85.0% and 4%, respectively. Furthermore, the results show that the antiviral activity is influenced by increasing the electron-withdrawing properties of substituents on the anilide ring and is positively influenced by increasing the lipophilicity; in this manner, the derivative showing values of R = 3-NO_2_ and log k = ca. 0.41 showed maximal activity with insignificant cytotoxicity. Di- and tri-substituted derivatives (R = 3,4-Cl, 3,4,5-Cl, 3-Cl-2-F, and 2,4-NO_2_) showed higher inhibition of H5N1 growth and simultaneous low cytotoxicity. For example, 3-Cl-2-F and 3,4,5-Cl exhibited virus growth inhibition activity values of 91.2 and 9.7% and cytotoxicity values of 79.3 and 2.4%, respectively; the log k values for these derivatives were 1.44 and 1.26, respectively. The study indicated that the antiviral activity linearly increases with increasing lipophilicity and is positively influenced by increasing the electron-withdrawing properties of substituents on the anilide ring. With the frantic search for drugs to treat COVID-19 patients and ultimately for a COVID-19 vaccine, the 8-HQ nucleus could play a role in this due to its promising antiviral activity; however, more research is required in this area.

## 3. Antibacterial Activity

The spread of the antibiotic-resistant bacterial strains constitutes a serious threat to public health. Some of these strains have even become resistant to many antibiotics and chemotherapeutic agents; hence, the term “multidrug resistance” has been introduced in the literature. Thus, the development of existing drugs and the discovery of new leads are major strategies used to combat the threat of these multidrug-resistant strains. In this context, Hu and coworkers prepared new derivatives of the known potent antibacterial agent 2,6-difluoro-3-hydroxybenzamide (DFMBA) [9]. The substituted 8-HQ **8** was alkylated with 1,3-dibromopropane in the presence of an aqueous sodium hydroxide solution and tetrabutylammonium iodide (TBAI) in dichloromethane to afford a mono-halogenated intermediate **9**. Target compound **10** was obtained via alkylation of DFMBA with the corresponding 8-HQ mono bromide (Figure 3).

The antibacterial activity of these derivatives was evaluated against several Gram-positive and Gram-negative bacteria using the zone-of-inhibition test. Inhibition zones were compared to the standard antibiotic PC190723 (**11**), which is an effective bactericidal cell division inhibitor that targets cell division protein (FtsZ) in several Gram-positive bacteria. The results showed that the antibacterial activities were lower than that of PC190723. In addition, the highest inhibition zone ratios (inhibition zone of the test compound/inhibition zone of the standard) of 0.25 and 0.18 against *S. aureus* were observed for derivatives where R = 5-Cl and 5,7-diCl, respectively. Moreover, when comparing the activity of 8-HQ derivatives with 8-HQ naphthalene analogues, the former were more active; this is probably due to the ring nitrogen, which may increase the polarity and water solubility of the compounds—factors that are required for antibacterial activity.

Krishna reported the synthesis of a series of new 5-amino-7-bromoquinolin-8-yl sulfonates **15** in good yield using a multistep synthesis method [10]. Bromination of 8-hydroxquinoline **1** with *N*-bromosuccinimide (NBS) in chloroform afforded 7-bromoquinolin-8-ol, which upon treatment with NaNO_2_/HCl followed by reduction with Na_2_S_2_O_4_ in 1:1 tetrahydrofuran (THF) and water gave 5-amino-7-bromoquinolin-8-ol **14**. The target sulfonate derivatives **15** were obtained from 5-amino-7-bromoquinolin-8-ol by reaction with various sulfonyl chlorides in dry THF in the presence of triethylamine (TEA) (Figure 4).

The newly synthesized sulfonate **15** derivatives were tested against several bacterial strains, such as *Staphylococcus aureus* and *Bacillus megaterium* (Gram-positive bacteria), *Klebsiella pneumoniae* and *Pseudomonas aeruginosa* (Gram-negative bacteria), and against two Gram-negative, antibiotic-resistant *E. coli* bacterial strains, namely mutant *E. coli* (Streptomycin-resistant) and donor *E. coli* (Rifampin-resistant), using the agar well diffusion method; amoxiclav (Ac) was used as the standard drug. Among the synthesized compounds, derivatives with an aryl group showed potent activity. For example, the compound with Ar = biphenyl exhibited potent activity against *Staphylococcus aureus*, with an inhibition zone of 22 mm compared to the reference drug (24 mm). On the other hand, derivatives with Ar = 4-FPh and 2-OH-5-NO_2_Ph displayed potent activity against *Pseudomonas aeruginosa*, with inhibition zones of 22 mm and 23 mm, respectively, compared to the standard drug (24 mm), whereas the derivative with R = 4-CH_3_Ph was potent against *Klebsiella pneumonia*, with an inhibition zone of 25 mm compared to the standard drug (27 mm).

A study published by Rbaa and coworkers described the synthesis of novel 4-aryl-3,4-dihydro-2*H*-[1,3]oxazino[5,6-*h*]quinolin-2-one (**17**) in high yields (65–90%). Refluxing a mixture of stoichiometric amounts of 8-hydroxyquinoline (8-HQ), *para*-alkylbenzaldehyde **16**, and urea under acidic conditions (37% HCl) for 48 h afforded **17** (Figure 5) [11].

These compounds were tested as antibacterial agents against six pathogenic strains, including *E. cloacae*, *E. coli*, *K. pneumoniae*, *P. aeruginosa*, *S. aureus*, and *A. baumanii*. Minimal inhibitory concentrations of these compounds were calculated and compared with three standard antibiotics, namely penicillin G, norfloxacin, and erythromycin, using the disk diffusion technique. Some of these prepared compounds exhibited remarkable antibacterial activity against Gram-positive and Gram-negative strains compared to the standard antibiotics used. One derivative (R = NO_2_, R′ = H) was more potent than the standard drugs and showed activity against *E. cloacae*, *K. pneumoniae*, *S. aureus, A. baumanii*, *E. coli*, *E. cloacae*, and *E. cloaca*, with minimum inhibitory concentration (MIC) values (mg/mL) of 1 × 10^−6^, 1 × 10^−5^, 1 × 10^−5^, 1 × 10^−4^, and 1 × 10^−4^, respectively, while the MIC for the standard drug was 1 × 10^−4^. Derivatives with R = R′ = H and R = H, R′ = Cl displayed similar activities against *S. aureus*, with MIC values of 1 × 10^−4^. Bioinformatic petra, osiris, and molinspiration (POM) analyses indicated that all compounds exhibit good bioavailability and less toxic profiles when compared with penicillin G.

Interestingly, two research groups have reported on the hybridization of 8-hydroxyquinoline with ciprofloxacin in a single step and evaluated the antibacterial activity of the hybrid molecule. Fu et al. reported the synthesis of 5-chloro-8-hydroxyquinoline-ciprofloxacin **19** via the Mannich reaction (Figure 6). Thus, 5-chloro-8-hydroxyquinoline **18** reacted with ciprofloxacin in the presence of paraformaldehyde in ethanol to afford the hybrid product at 75% yield [12].

The hybrid **19** was screened against both Gram-positive and Gram-negative (*Staphylococcus epidermidis*, *S. aureus*, *Enterococci faecalis*, and *E. faecium*) bacterial strains, and the minimum inhibitory concentrations (MICs) were determined via the agar dilution method. The results indicated that the hybrid shows exciting promise against both Gram-positive and Gram-negative bacteria. It displayed significant effects against both susceptible and drug-resistant strains, with MIC values of 4–16 µg/mL. These values were lower than those of standard drug (ciprofloxacin), which has MIC values of 0.125–0.5 µg/mL. However, the authors speculated that the introduction of a quinolone skeleton might be an effective way to modify these kinds of compounds into a novel class of broad-spectrum antibacterial agents with a specific mode of action.

On the other hand, Vu and coworkers prepared a hybrid compound **21** at 96% yield through coupling of 8-hydroxyquinoline-2-carboxylic acid **20** and ciprofloxacin by using 2-(1*H*-benzotriazole-1-yl)-1,1,3,3-tetramethylaminium tetrafluoroborate (TBTU) and *N*,*N*-diisopropylethylamine (DIEA) (Figure 7) [13].

The hybrid was screened for cell toxicity against HeLa cells, a tumor cell line, primary cell cultures of mouse fibroblasts, and against non-cancerous mammalian cell line. It showed no cell toxicity at the concentrations tested (0–200 µM). Then, the antibacterial activity was tested against Gram-negative and Gram-positive bacteria, including pathogenic bacteria present in the hospital environment that are difficult to treat (*Enterococcus faecium*, *Staphylococcus aureus*, *Klebsiella pneumoniae*, *Acinetobacter baumannii*, *Pseudomonas aeruginosa*, and *Enterobacter species*). The results indicated that that hybrid **21** exhibits more potent activity than the standard drug against *Staphylococcus aureus*, with an MIC (mg/mL) value of 0.0625 (MIC of the standard drug = 0.125). However, the hybrid was less active against the rest of the pathogens as compared to the standard drug. In general, the two aforementioned hybrids were systematically less active than the parent antibiotic, ciprofloxacin.

From these published data, it is obvious that 8-HQ could be an important motif for future antibacterial drugs, especially against antibiotic-resistant strains. However, more research is required, including experiments with animals using newly synthesized antibacterial agents.

## 4. Antifungal Activity

Fungal resistance to antifungal agents has become a problem and a challenge to scientists in recent years. This resistance has serious implications for morbidity, mortality, and health care costs worldwide. Hence, attention has been given to understanding the mechanism of antifungal resistance in order to develop new treatments and strategies to manage infections caused by resistant organisms. Along this line, Le Pan and colleagues reported the synthesis of conjugates of 7-hydroxycoumarin (umbelliferone) with several heterocylic rings, including 4- and 8-hydroxyquinolines [14]. 7-Hydroxycoumarin was mono-alkylated in moderate yields via an excess of Br-(CH_2_)_n_-Br (*n* = 2,4) in refluxing acetone in the presence of K_2_CO_3_-NaOH (2:1). Bromoalkylated coumarins were then reacted with 4- and 8-hydroxyquinoline in the presence of 1 equivalent of KOH and catalytic amounts of KI and tetra-*n*-butylammonium bromide (TBAB) as a phase transfer catalyst in refluxing acetonitrile to give the corresponding conjugates at 55–80% yields (Figure 8).

The antifungal activity of these new conjugates against four phytopathogenic fungi (*A. alternate*, *A. solani*, *B. cinerea*, and *F. oxysporum*) was evaluated by measuring the mycelial inhibition of radial growth on potato dextrose agar (PDA) media compared to the commercial fungicide carbendazim. The inhibition percentages of radial growth against all of these fungi for umbelliferone-8-hydroxyquinoline (*n* = 2), umbelliferone–8-hydroxyquinoline (*n* = 4), and umbelliferone-4-hydroxyquinoline (*n* = 4) were 49–63, 57–90, and 18–41%, respectively, compared to carbendazim (95–99%) at 200 µg/mL concentration. When these derivatives were tested at a series of lower concentrations to measure the half-maximal effective concentration (µg/mL), umbelliferone-8-hydroxyquinoline (*n* = 4) was the most active compared to the reference drug at 10.6 and 2.3 µg/mL, respectively. The rest of the 8-hydroxyquinoline derivatives exhibited weak activity. The results also showed that umbelliferone derivatives with a spacer of 4 carbon atoms exhibit better antifungal activity than those with a spacer of 2 carbon atoms. The chain length may contribute to the activity by influencing the flexibility of the molecules. In addition, the position of the OH group affects the antifungal activity; derivatives with OH at position 8 were more active than those with OH at position 4. The structure–activity relationship suggests that modification of the umbelliferone-8-hysroxyquinoline analogues could help to develop highly selective and low phytotoxic fungicides. The phytotoxicity of effective compounds was evaluated at 200 mg/L with *L. sativa*, with the results showing that umbelliferone-8-hysroxyquinoline (*n* = 4) had no phytotoxic effects on the seedling growth of lettuce.

Yurras and coworkers have reported on the multistep synthesis of novel 3,4,5-trisubstituted triazole derivatives bearing 8-hydroxyquinoline **31**, evaluating their antimicrobial activity [15]. Synthesis was achieved by first reacting 8-hydroxyquinoline **1** with ethyl 2-chloroacetate in refluxing acetone in the presence of a base. Treatment of the formed ester **26** with hydrazine hydrate in ethanol followed by phenyl isocyanate afforded the corresponding thiourea **28**. Ring closure using KOH in refluxing ethanol led to the formation of 3-mercapto-1,2,4-triazole derivative **29**. Reaction of 3-mercapto-1,2,4-triazole with 2-chloro-*N*-(substituted (benzo)/thiazole)acetamide **30** afforded the target compounds, as depicted in Figure 9.

The newly synthesized triazole-8-hydroxuquinoline derivatives were tested against four fungi—*Candida glabrata* (ATCC 90030), *Candida parapsilosis* (ATCC 22019), *Candida albicans* (ATCC 90028), and *Candida krusei* (ATCC 6258)—using fluconazole as a standard reference drug. The results indicated that these compounds exhibited low activity against fungi where MIC values ranged between 31.25 and 1000 mg/mL, while the standard drug had an MIC range of 1.95–62.5 mg/mL against four Candida species.

Some sulfonates reported by Krishna (Figure 4) were also tested in vitro against *Aspergillus niger* and *Penicillium spinulosum* fungal strains by the poison plate technique; fluconazole was used as a standard drug for antifungal activity. Two compounds with Ar = biphenyl and Ar = 2-nitro-5-hydroxybenzene exhibited the most potent antifungal activity against *Aspergillus niger*, with inhibition zones of 10 and 13 mm, respectively, compared to fluconazole (15 mm); and against *Penicillium spinulosum*, with inhibition zones of 12 and 10 mm, respectively, compared to fluconazole (12 mm).

## 5. Anticancer Activity

Cancer is a dreadful disease that has become a global burden, causing thousands of deaths per year, despite the technological and pharmaceutical improvements over the past several years. It has emerged as one of the leading causes of mortality worldwide [16]. Cancer treatments include surgery, radiotherapy, and anticancer drugs (chemotherapy), in addition to other specialized techniques. Much scientific effort is being made every day in the fight against cancer, but successful treatment of some cancer types is still a challenge that needs more work [17]. On the synthetic front in the fight against cancer, two new series of derivatives have been prepared where the bioactive quinolone motif is incorporated, as shown in Figure 10 [18]. 6-Bromo-8-methoxyquinoline (**1**) was prepared according to a published procedure [19]. Afterwards, compound **32** was subjected to a Suzuki cross-coupling reaction with *p*-formphenylboronic (**33**) acid to afford synthon **34** in excellent yield. All target products (**35**) were synthesized via a simple and effective method using a one-pot Mannich-type reaction that involves a reaction of amines, carbonyl compounds, and dialkylphosphonate.

The cytotoxicity of these prepared compounds against esophageal (Eca109) and hepatocellular (Huh7) cancer cell lines was evaluated using sunitinib as a positive control. The results showed that most of these compounds exhibit moderate to high activity, with IC_50_ values ranging from 2.26 to 7.46 μmol/L for the two most promising derivatives containing 2-methylphenyl and 4-methylphenyl groups for R^2^ and *iso*-propyl for R^1^; some of these compounds exhibited inhibition activities comparable to those of sunitinib, which showed IC_50_ values of 16.54 and 5.27 μmol/L towards Eca109 and Huh7, respectively. Furthermore, the results indicated that ethyl and isopropyl substituents of phosphonate have no major effects on the cytotoxicity activity, while substituents on the phenyl ring showed significant influence on the bioactivity.

Faydy and coworkers described the synthesis of new derivatives of 8-hydroxyquinoline (**36**–**38**) (Figure 2) in a multistep approach [20] according to published procedures [21,22,23]. The antioxidant activity of prepared products, along with L-ascorbic acid, was evaluated by means of the free radical scavenging method using the 2,2-diphenyl-1-picryhydrazyl (DPPH) assay. The results revealed that all products showed low antioxidant activity, with IC_50_ values of 0.8–2.49 mg/mL, whereas the IC_50_ value of 1-ascorbic acid was 0.1 mg/mL. Furthermore, the results showed that as the number of hydroxyl groups increases, the inhibition activity increases too.

A recent paper by Shah et al. (2018) discussed the preparation of a series of 8-hydroxyquinoline hydrazones (**39**) at C2 (Figure 3) in moderate to excellent yields [24] following a published procedure [25]. The structures of prepared compounds were confirmed with the aid of a panel of spectroscopic methods, including ^1^H NMR, ^13^C NMR, IR, and HRMS.

The prepared compounds were then subjected to cell viability evaluation against Hela, MCF-7, A-549, and MDA-MB-231 (triple-negative breast cancer cell line) cell lines using the MTT assay [26,27,28] at a concentration of 20 mM. The results revealed that only 4 compounds designated with R = phenyl, 3,5-dimethylphenyl, 4-fluorophenyl, and 4-trifluoromethylphenyl exhibited significantly reduced cell viability percentages towards the tested cancer cell lines. The IC_50_ values for these compounds were between 26.30 and 63.75 mM against selected cancer cell lines, whereas the IC_50_ of docetaxel (positive control) was in the range of 3.37–4.46 mM.

A new series of glycoconjugates composed of various sugar units (**40**, **41**) (d-glucose or d-galactose) and 8-hydroxyquinolines (**42**, **43**) was prepared in an effective and simple method [29]. The connection between these units was accomplished by *O*-glycosidic bond or via *O*-methylene 1,2,3-triazole linker, as shown in Figure 11. Sugar derivatives of **44** and **45** were prepared according to published procedures [30,31,32]; sugar was used to enhance the bioavailability and solubility of potential drugs.

All prepared derivatives were tested against different cancer cell lines, including Hela, HCT 116, and MCF-7, in addition to a normal human dermal neonatal fibroblast (NHDF-Neo). Compound **2**, designated with R = R_1_ = H and R_2_ = OAc, was the most active, with IC_50_ values of 30.98, 22.7, and 4.12 mM against Hela, HCT 116, and MCF-7, respectively. Other derivatives exhibited low bioactivity. This could be attributed to the use of the quinolone hydroxyl group to form a glycosidic linkage, which impedes the chelation of metal ions due to steric hindrance. In this respect, it should be stated that the presence of the 1,2,3-triazole moiety improves the activity of glycoconjugates, whereas the type of sugar fragment did not affect the activity significantly. Finally, conjugates with a sugar moiety and free hydroxyl group exerted better inhibitory potential than acetylated analogs.

Fouda published a paper dealing with the synthesis, characterization, and cytotoxicity of new derivatives of halogenated 2-amino-4-aryl-4-pyrano[3,2-*h*]quinolone-3-carbonitrile (**48**) derivatives (Figure 12) [33]. These derivatives were prepared through interactions of various 8-hydroxyquinolines (**46**) with α-cyanocinnamonitriles (**47**).

These prepared compounds were screened for potential anticancer activity against MCF-7, HCT 116, HepG-2, and A549 using the MMT assay. Structure–activity relationship (SAR) results revealed that 6-chloroanalogues were the most active, whereas the 9-methylanaloguess were the least potent. In addition, the lipophilicity of the products increased in the presence of halogen atom substituents at positions 4, 6, and 9. The IC_50_ values (mg/mL) of these compounds were in the ranges of 0.9–38.2, 1.3–45.5, 0.7–44.5, and 1.23–36.7 against MCF-7, HCT 116, HepG2, and A549, respectively; while the colchicine reference showed IC_50_ values of 6.1, 2.6, 4.6, and 3.78 mg/mL, respectively.

In a similar fashion, Chhabra et al. (2017) described an Amberlite IRA 402(OH)-mediated synthesis of novel benzothiazole–quinoline conjugates with excellent yields [34]. A synthetic procedure (Figure 13) involved a condensation reaction between the amino group (NH_2_) in **49** and the carbonyl group in salicylic aldehyde (**50**), followed by intramolecular cyclization under a microwave approach to form the benzothiazole (**51**). The reaction between **51** and 5,7-dialkyl-8-hydroxyquinoline as a phenolate ion in the presence of a catalytic amount of Amberlite IRA and an excess amount of dibromoalkane under microwave conditions afforded the target product **52**; dibromoalkanes were employed as linkers between the two fragments, 8-hydroxyquinoline and 2-benzothiazol-2-ylphenol. The prepared compounds were then subjected to a cytotoxicity study along with cisplatin (a positive control) against a panel of cancer cell lines, including HeLa, MCF-7, A549, and human ovarian carcinoma (A2780), using the MMT assay. Most of the prepared compounds were more potent than cisplatin, with IC_50_ values in the ranges of 5–19, 7–49, 10–30, and 10–38 mM against MCF-7, HeLa cells, A2780, and A549, respectively. In addition, the target products were 3–25-fold more selective in cancer cell lines than normal fibroblasts.

Gayathri and coworkers prepared a novel compound (Figure 4) bearing three quinolinone moieties in a simple procedure that involved a reaction of 3,6-bis(bromomethyl)-2-chloroquinoline and 8-hydroxyquinoline at a ratio of 1:2 in acetone (aprotic polar solvent) under reflux conditions. The target product was fully characterized by using various spectroscopic techniques and single-crystal X-ray diffraction method. This compound was screened against MCF-7 and Hela cancer cell lines using MMT assay, giving IC_50_ values of 21.02 and 27.73 mM, respectively [35].

In addition, Shamsi and coworkers prepared 16 quinoline-based 1,3,4-oxadiazole-triazole derivatives (Figure 14) based on the hybrid strategy of nitrogen-containing heterocyclic scaffolds [36]. These compounds were considered to be high-impact motifs with a wide range of biological activities [37,38]. The synthetic strategy was based on the treatment of 1 with carbonate to produce the phenolate anion, which reacts with ethyl chloroacetate under S_N_2 conditions to afford the corresponding intermediate. Reaction of this intermediate with hydrazine produced 54, which undergoes intramolecular cyclization in the presence of carbon disulfide and the base (KOH) to give compound 55 (1,3,4-oxadiazole) after acidification. The thiol group (Ar-SH) in 55 is acidic and gives the Ar-S^−^ anion upon treatment with a base. Then, the corresponding anion reacts with propargyl bromide to afford compound 56. Finally, reaction of various azide derivatives of 56 yielded the target derivative 57 through a (3 + 2) cycloaddition mechanism.

The anticancer activities of all of these derivatives were examined against four different human cancel cell lines, namely human lung carcinoma (A-549), hepatocellular carcinoma (HepG2), human cervical carcinoma-HPV18 (Hela), and human cervical carcinoma-HPV16 (SiHa), using the MMT colorimetric assay; normal cells were used as controls, whereas doxorubicin was used as the reference drug. The results indicated that the product with an *o*-chloro substitution on the phenyl ring was the most potent, with an IC_50_ value of 5.6 mM against the A-549 cell line, which is higher than that of doxorubicin (IC_50_ = 1.83 mM). Interestingly, this compound was not toxic towards normal cells (up to 200 mM concentration).

A series of styrylquinolines with various substituents was prepared as shown in Figure 15 [39]. In the first step of the synthesis, the hydroxyl group in **58** was protected by conversion into the acetyl analogue **59**. Then, a condensation reaction between a methyl group at position 2 of protected 8-hydroxyquinoline and appropriate aromatic aldehydes as carried out using microwave heating or conventional procedures [40]. Finally, deprotection of the acyl group was achieved with carbonate anion–methanol or pyridine–water mixture afforded the target products **60**.

All prepared compounds were screened for anticancer activity against the wild-type HCT 116 P53^+/+^ and HCT 116 P53^−/−^ cells and for cytotoxicity against normal cell fibroblasts. The results revealed that derivatives that have hydroxyl or acyloxy groups at position 8 of the quinolone moiety exhibit moderate activities (4.60–25.00 mM) and (2.61–25.00 mM) towards P53^+/+^ and P53^−/−^, respectively. Analogues that were based on dichloroquinone and oxyacyl groups were the most active in this series towards P53^+/+^ and P53^−/−^ (0.28–13.85 mM and 0.27–12.15 mM, respectively). On the other hand, dichloro-8-hydroxyquinone derivatives showed IC_50_ values in the ranges of 0.73–10.48 mM and 0.54–15.30 mM towards P53^+/+^ and P53^−/−^, respectively. It should be stated that 5-fluorouracil and doxorubicin have IC_50_ values of around 4.5 and 0.35 mM against P53^+/+^ and P53^−/−^, respectively. Apparently, the presence of strong electron-withdrawing substituents (Cl, NO_2_, CN) in the styryl moiety is critical for high anticancer activity.

Schmitt et al. (2019) prepared a series of 4-aryl-pyrano[3,2-*h*]quinolines in yields ranging from 30 to 60% [41]. Their synthetic procedure involved a Knoevenagel condensation between malonitrile (**61**) and the respective aryl aldehyde **62** to afford the corresponding arylidene–malonitrile intermediates **63**, as shown in Figure 16. Then, the phenolate anion attacks the β-carbon via C-7 to produce an acyclic Michael adduct, which undergoes cyclization reaction (6-exo-dig) and tautomerization to afford the target product **64**. All prepared derivatives in the study (along with compound LY290181 used as the standard) have been tested against various cancer cell lines, namely pancreatic carcinoma 518A, colon carcinoma cells HT-29, DLD-1, HCT 116, cervix carcinoma KB-V1^Vbl^, and MCF-7^Tobo^ breast carcinoma cell lines, in addition to non-malignant fibroblasts. The results revealed that that these derivatives exhibit remarkable activities, with IC_50_ values in nanomolar concentrations, meaning these results are even better than the activity of LY290181. Two compounds designated with R = CH_3_, R_1_ = R_3_ = H, R_2_ = R_4_ = F and R = CH_3_, R_1_ = R_3_ = R_4_ = H, R_2_ = NO_2_ were the most active among all examined molecules. The first one had IC_50_ values of 20.1 and 14 nM against MCF-7 and KB-V1^Vbl^, respectively. On the other hand, the second compound had an IC_50_ value of 20 nM against KB-V1^Vbl^, which is even better than the reference. In this regard, it is worth mentioning that the mechanism of action of these compounds may be associated with tubulin polymerization interference and ROS formation, in which the molecule-induced ROS generation could be responsible for their cytotoxicity, since ROS overproduction may induce endoplasmic reticulum stress.

Matrix metalloproteinases (MMPs) play significant roles in cancer diseases, with MMP-2 and MMP-9 being important types among the various MMPs. For instance, they could induce the release of cell membrane precursors of growth factors (e.g., epidermal growth factor receptor) ligands, which promote tumor proliferation [42,43]. Along this line, Chen and colleagues described the synthesis of two series of 8-hydroxyquinolines, as shown in Figure 17 and Figure 18 [44]. In Figure 17, the first step involved protecting the amino group in **65** with *tert*-butyloxycarbonyl (Boc), then the free carboxylic acid group in **66** reacted with the amino groups in various substrates leading, to the formation of carboxamide **67**. Other steps involved deprotection to liberate the free primary amine **68** [45], which reacts with 5-chloro-8-hydroxyquinoline-7-carboaldehyde or 8-hydroxycarbaldehyde through reduction amination to yield target product **69**. Furthermore, in Figure 18, the phenolic hydroxyl group in **70** was protected and reaction of the primary amino group in **70** with substituted carboxylic acids afforded derivative **71**. Finally, deprotection of the hydroxyl group yielded the target derivative **72**.

All of these derivatives were screened as potential MMP-2/9 inhibitors. The results revealed that compounds (of series 1) that have substituents at C-7 on the quinolone moiety showed IC_50_ values (mM) in the range of 0.81–10, while for those with substituents at C-5, the IC_50_ values ranged from 5.7 to 10. On the other hand, derivatives belonging to series 2 showed IC_50_ values in the range of 6.5–10 against MMP-2. As for MMP-9, using the same sequence, the IC_50_ values (mM) were in the ranges of 1.3–10, 5.1–10, and >10. Some selected compounds (those with substituents at C-7) were further screened against a panel of cancer cell lines (HL60, K562, KG1, A549, PC-3, and MCF-7), along with human umbilical vein endothelial cells. The results indicated that most of the tested compounds exhibited good bioactivity, with IC_50_ values in the range of 0.69–22 mM. Finally, the positive control, the hydroxamate-based MMP inhibitor NNGH [46], showed IC_50_ values of 29–187 mM for antiproliferation activities against various cancer cell lines, and against MMP-2 and MMP-9 showed values of 0.0091 and 0.0088 mM, respectively.

Ökten and coworkers described the synthesis of 5,7-dibromo-8-hydroxyquinoline [47] in excellent yield via reaction of 8-hydroxyquinoline with two equivalents of bromine in chloroform. The target molecule exhibited IC_50_ values (mg/mL) of 5.8, 17.6, 18.7, 5.4, 16.5, and >1000 against A549, FL, HeLa, HT29, MCF7, and Hep3B, respectively. From all of these studies, one can see the importance of the 8-HQ moiety in potential anticancer drugs. In addition, some of the prepared compounds could be leads towards the development of potent and safe drugs. However, more work is required in this category, which could involve the use of animals and possibly human subjects to evaluate the efficacy and safety profiles of the prepared compounds.

## 6. Alzheimer’s Disease

Alzheimer’s disease (AD), characterized by a loss of cognitive ability and severe behavioral irregularities, is a chronic neurodegenerative disorder. It is most common among the elderly, and can be described as an irreversible brain disorder that breaks down memory and reduces the ability of a patient to carry out simple mental and cognitive functions, such as comprehension, solving simple problems, and trivial calculations. This disease is becoming a universal health problem, and it can eventually lead to death [48]. Statistics indicate the presence of approximately 2.5 to 4.0 million Alzheimer’s disease patients in the United States, and 17 and 25 million worldwide [49]. Published research findings indicate that cholinergic dysfunction could be associated with selective and irreversible deficiency of the neurotransmitter acetylcholine, which is controlled by hydrolysis of acetylcholine via acetylcholinestrase (AChE) and butyrylcholinestrase (BChE). Additionally, it was suggested that AChE predominates in a healthy brain, whereas BChE is considered to play a minor role in regulating the brain’s ACh levels [49].

The approved prescribed commercial drugs for the treatment of AD, which provide slight improvements in memory, include donepezil, rivastigmine, and others; their action is based on the inhibition of acetylcholinesterase. It is also worth mentioning that other factors contribute to AD, such as β-amyloid (A β) deposits and oxidative stress. In this respect, several studies have shown that levels of redox-active metal ions, including Cu^2+^ and Zn^2+^, are observed in the brains of AD patients [50]. These metal ions can interact with β-amyloid peptides to form insoluble plaques [51,52].

In the search for potent BuChE and AChE inhibitors, Hirbod et al. (2017) designed and prepared eight novel compounds incorporating coumarin and 8-hydroxyquinoline moieties (**73**), as shown in the [53]. These researchers used various dibromoalkanes (*n* = 3–5) as cross-linkers between 8-hydroxyquinoline and coumarin rings in the presence of an aprotic solvent (*N*,*N*-dimethylformamide). In addition, the activity levels of these prepared compounds were evaluated against BuChE and AChE using Ellman’s method. The results demonstrated that some of the prepared compounds exhibited potent AChE and BuCHE inhibition activities, with half-maximal inhibitory concentration (IC_50_) values ranging from 8.80 to 26.50 µM, respectively. The IC_50_ values of the commercially used drug donepezil were 0.016 and 5.41 mM, respectively.

In a similar fashion, Yang et al. (2018) prepared a series of multitargeted 8-hydroxyquinoline derivatives (**74** and **75**), as shown in Figure 5 [54]. The target compounds were prepared by chloromethylation of 8-HQ to give 5-chloromethyl-8-hydroxyquinoline. Then, the former compound was reacted with *tert*-butylpiperazine-1-carboxylate followed by trifluoroacetic acid to remove the protected Boc group. Finally, the resulting compound was reacted with the appropriate cinnamic or hydroxycinnamic acids, using 1-ethyl-3-(3-dimethylaminopropyl)carbodiimide (EDC). In this context, β-amyloid (Aβ) significantly contributed to the progression of Alzheimer’s disease (AD), where elevated levels of Aβ have been detected in the brains of AD patients. Moreover, Aβ *A* aggregation can be clearly observed in the presence of Cu^2+^, Zn^2+^, and Fe^2+^ ions, since these ions can readily bind to Aβ through some of its specific residues. In addition, Aβ can aggregate by itself. Th prepared target compounds were tested for their inhibition of Aβ aggregation using the thioflavon T-binding assay [55,56]. Furthermore, chelating studies of Cu^2+^, Zn^2+^, Fe^3+^, and Fe^2+^ were conducted using the former prepared derivatives by means of a UV-Vis spectrophotometer. The results revealed that the compound designated with R_1_ = H, R_2_, R_3_ = OCH_3_ showed the maximum percentage inhibition of Aβ_1→42_ aggregation of 65.82% with an IC_50_ of 5.64 mM compared to resveratrol, which showed corresponding values of 51.74% and 12.43 mM. In addition, the previous compound was selected as a representative to examine its metal chelation behavior; it exhibited significant activity in this context, producing even better results than clioquinol.

Hu et al. (2019) discussed the design and synthesis of a series of 8-hydroxyquinoline derivatives, as shown in Figure 19, Figure 20 and Figure 21 [57]. Synthesis of these compounds involved protection of the phenolic group in **76** using Boc group to allow the reaction of the primary aromatic amine in **77** with substituted benzoyl chlorides in the presence of triethylamine (TEA) as the base and catalyst to form the carboxamide group **78**. In the final step, deprotection of the Boc group was accomplished with hydrogen chloride to afford the desired products as salt **79**. Regarding Figure 20, the phenoxide anion in **80** was formed in situ by reaction of the phenolic group with potassium carbonate (base), which then reacts with benzyl bromide. The methyl group in **81** was then subjected to Pinnick oxidation using SeO_2_ to afford the corresponding acid **82**. Reaction of the carboxylic acid group in **82** with thionyl chloride and then with 3-(cyclopentyloxy)-4-methoxyaniline or 3,4-dimethoxyaniline afforded **83** upon removal of Bn with hydrogen gas in the presence of the palladium catalyst yielded the desired product **84**. On the other hand, synthesis of compound **87** was achieved by converting **1** into *tert*-butyldimethylsilyl (TBS)-protected 2-aminoquinolin-8-ol (**85**). This process was carried out by oxidation of the starting material using *m*-chloroperbenzoic acid (mCPBA) to form *N*-oxide, which was subsequently followed by refluxing with dimethylsulfate, treatment with NH_4_OH, and finally protection with *tert*-butyldimethylsilyl chloride (TBSCl). The obtained derivative **86** was treated with the corresponding benzoyl chlorides, then went through the deprotection process using tetrabutylammoniumfluoride (TBAF) to afford the phenolic group in the target derivative **87** (Figure 21).

Compounds **79**, **84**, and **87** are considered hybrids of clioquinol–rolipram and roflumilast as multitarget-directed ligands for the treatment of AD in terms of inhibition of phosphodiesterase 4D (PDE4D), the oxygen radical absorbance capacity (ORAC) value, and the experimental potential of the blood–brain barrier (BBB) permeability (*Pe*) of the selected compounds using parallel artificial membrane permeation assay (PAMPA). In this respect, PDE4D is involved in the process of long-term potentiation and memory consolidation. One of the derivatives of **79**, for which X = H, R^1^ = CF_2_H, R^2^ = cyclopentylethyl, exhibited an IC_50_ of 0.399 mM against PDE4D compared to rolipram as the reference compound (IC_50_ 0.621 mM), whereas the derivative showing X = H, R^1^ = cyclopentylmethyl of family **84** exhibited the highest value among all tested compounds in the ORAC assay (its value is 1.98 expressed as Trolox equivalents). In this respect, the ORAC value of antioxidant activities increases the ability of a given compound as the antioxidant becomes better [58]. Clioquinol, rolipram, and roflumilast showed values of 0.60, 0.070, and 0.067, respectively. Oxidative stress has a major effect in the production of excess free radicals, which lead to cell death and cytosceletal damage in AD [59]. Finally, the BBB has a major role in the generation of chronic brain inflammation during AD [60]. The *Pe* values of selected compounds were evaluated using PAMPA. The results indicated that a derivative of **87** with R^1^ = CH_3_ and R^2^ = cyclopentylmethyl exhibited a maximum value of *Pe* 16.41 (±0.44) × 10^6^ cm s^−1^, whereas other tested compounds showed values higher than 4.7 × 10^6^ cm s^−1^, indicating that these compounds may cross the BBB. The *Pe* values for rolipram, roflumilast, and clioquinol were evaluated as (18.87 ± 0.57), (9.22 ± 0.62), and (5.20 ± 0.33) × 10^6^ cm s^−1^, respectively.

An earlier paper by Wang et al. (2018) described the synthesis of new 8-hydroxyquinoline derivatives, as shown in Figure 6 [61]. Compounds **88** and **89** were prepared from 8-HQ by reaction first with formaldehyde in the presence of hydrogen chloride, followed by reaction with triethylphosphite to afford diethyl ((8-hydroxyquinolin-5-yl)methyl)phosphonate [56]. In this reaction, the phenolic group was protected via reaction with chloro(methoxy)methane to yield diethyl ((8-(methoxymethoxy)quinolin-5-yl)methyl)-phosphonate. Subsequent reaction of the last compound with various 2-nitrobenzaldehydes in the presence of sodium hydride yielded derivatives of 8-(methoxymethoxy)-5-(2-nitrostyryl)quinolone. Reductive cyclization of 2-nitrostyrenes with carbon monoxide followed by removal of the protecting group (methoxymethane) by hydrochloride solution led to the formation of the final product. For compound **89**, diethyl ((7-chloro-8-hydroxyquinolin-5-yl)methyl)-phosphonate was prepared by reacting diethyl ((8-hydroxyquinolin-5-yl)methyl)phosphonate with sodium hypochlorite, followed by the same previous steps. On the other hand, compound **91** was prepared through a series of steps that involved reacting 2-methyl-8-hydroxyqunolines **90** with acetic anhydride, then with 2-nitrobenzaldehydes, followed by reductive cyclization of 2-nitrostyerenes in the presence of the palladium (II) acetate catalyst under carbon monoxide gas, and finally with a base (carbonate or methoxide ions, depending on the nature of X).

Compounds **88, 89**, and **91** were tested against the ORAC assay, BBB permeability assay, and inhibition activity towards amyloid beta (*Aβ*) self-induced aggregation. The results revealed that compounds **88** (R = OH), **89** (x = H, R_1_ = OH, R_2_ = H), and **91** (x = H, R_1_ = OH, R_2_ = CH_3_) exhibited the highest activities among all prepared compounds (6.6, 5.3, 5.9, and 5.4, respectively). The presence of the phenolic hydroxyl group at C-5 of the indole moiety enhances the activity. On the other hand, the presence of a chlorine atom in compound **89** lowers the activity compared to a hydrogen. The reference standards, including clioquinol, melatonin, and a mixture of clioquinol and melatonin, exhibit the values of 0.5, 2.4, and 2.9, respectively. This test was performed based on a fluorescein (ORAC-FL) method with a Trolox as the internal standard. Similarly, permeation of the BBB is considered an important parameter for potential central nervous system (CNS) candidates; this assay was accomplished using the PARMA method. The results showed that most of the target products could effectively permeate the BBB through passive diffusion. Two compounds of series **91**, designated as x = H, R_1_ = OH, R_2_ = H and x = H, R_1_ = OH, R_2_ = OCH_3_, showed higher BBB permeability activity (*Pe* values 14.8 and 12.1, respectively) compared to other derivatives that have a hydrogen atom instead of the phenolic hydroxyl group. In contrast, compounds **88** and **89**, bearing the indole moiety at position 5 of the quinolone scaffold, gave *Pe* values of 9.1 and 6.8, respectively, even in the presence of a phenolic hydroxyl group.

*A*β represents the major component of amyloid plaques found in the brains of Alzheimer’s patients [62]. Inhibition of *A*β self-induced aggregation for the prepared 8-hydroxyquinoline-indole derivatives was examined using thioflavin fluorescence. One of the derivatives of **91**, for which X = H, R^1^ = OH, R^2^ = H, caused 51.2% percent inhibition, which was the highest percentage among the prepared compounds. On the other hand, compounds **88** and **89** caused 57.2% and 68.7% inhibition, respectively; whereas clioquinol, curcumin, and resveratrol showed 1.9%, 36.7%, and 42.1% inhibition, respectively.

In a recent publication, Prati and coworkers [63] described the synthesis of a new series of 8-hydroxyquinoline derivatives (**94**) for which the products possess structural features of two commercial drugs, namely donepezil and clioquinol. Depicted in Figure 22 are the steps involved in the synthesis of these compounds.

This synthesis was a multicomponent Mannich reaction that involved a mixture containing piperazine (**92**), paraformaldehyde, and 8-HQ or its 5-chloroanalogue under a microwave-assisted procedure to afford 7-(piperazin-1-ylmethyl)-8-hydroxyquinolines (**93**). This synthon was then reacted with various benzyl chlorides in DMF. Compound **94** and its derivatives were assayed for potential inhibitory activity towards human anti-hAChE and anti-hBChE. The results indicated that at a concentration of 40 mM, all derivatives exhibited inhibition, with values ranging from 9.0 to 63.8% and 49.2 to 89.1% for 5-chloro-8-hydroxyquinoline and 8-hydroxyquinoline derivatives, respectively, against hBChE. However, these compounds were inactive or showed very weak inhibition activity against hAChE, suggesting selectivity of the target products towards hBChE. In addition, these results highlighted the effect of the chlorine atom at position 5 of the 8-hydroxyquinoline moiety on the activity compared to the hydrogen atom. The chemical references donepezil, tacrine, and galantamine exhibited inhibition towards hBChE, with values of 84.3, >90, and 65.8%, respectively.

On the other hand, the inhibition activity of compound **10** and its derivatives towards the Aβ_42_ antiaggregating property was evaluated. The obtained results demonstrated that the inhibition potencies of all derivatives ranged from 19.1 to 65.0% at the concentration of 50 mM; both series (X = H, Cl) had close activity rates. In addition, the metal chelating ability of the selected compounds was examined using Cu^2+^ and Zn^2+^ in phosphate buffer solution (pH 7.4). Spectroscopic data showed that there is a bathochromic shift of about 18 nm from the original band (243 nm) upon complex formation; as the metal ion concentration increases (1.56 to 50 mM), the intensity of the absorption band also increases. Finally, all derivatives showed antioxidant activity, whereby some prepared compounds exhibited higher activity than Trolox.

## 7. New Chemistry

Raj and Padhi reported that the condensation of 8-hydroxyquinoline-2-carbaldehyde (**95**) with aromatic diamines (**96**) afforded quinoline-based benzimidazole (**97**) followed by intracyclization reaction to produce derivatives of **100**. With aliphatic diamines (**98**), compound **95** produced *bis*-imines (**99**) without undergoing intracyclization reaction, as shown in Figure 23 [64]. Products of these reactions have been well-characterized using various techniques, including FT-IR, NMR, MS, and single-crystal X-ray diffraction method. In the first step of both reactions, one equivalent of 8-hydroxyquinoline-2-carbaldehyde underwent a condensation reaction with one equivalent of primary amine to form the mono-imine product (Schiff bases). However, in the second step and with the presence of aromatic amine, an intramolecular ring cyclization occurred, where the resulting precursor reacts further with the second equivalent of 8-hydroxyquinoline-2-carbaldehyde, followed by migration of hydride to afford compound **100**. In the case of an aliphatic amine, the second step represents a second condensation reaction with another molecule of 8-hydroxyquinoline-2-carbaldehyde to give the final product **99**.

Evtushak and Vorob’ev described the synthesis of 9-hydroxypyrazolo[1,5-a] quinolones and 2-substituted-9-hydroxy[1,2,4] triazolo[1,5-a]quinolones, as shown in Figure 24 [65]. Concerning the former derivatives, the *N*-amination of **1** was achieved by using *O*-(mesitylenesulfonyl)hydroxylamine (MSH) as an *N*-aminating agent [66] to produce 1-amino-8-hydroxyquinolonium mesitylenesulfonate (**101**). Compound **101** then reacted with alkenes or alkynes containing electron-withdrawing substituents under the mechanism of 1,3-dipolarcycloaddition and in the presence of a base (K_2_CO_3_) to afford **102**. On the other hand, the reaction of **101** with aromatic nitriles or acetonitrile in aqueous potassium hydroxide solution yielded 1,2,4-triazole-based quinolone (**103**).

On the other hand, Kong et al. (2017) prepared novel barbituroquinoline derivatives **106** and **107** in a one-pot procedure by combining three components in water, namely **1**, 2-thiobarbituric acid (**104**), and aldehyde (isatin) **105** [67], as shown in Figure 25. This reaction was conducted under mild experimental conditions and without a catalyst.

## 8. Conclusions and Future Directions

The 8-hydroxyquinoline moiety can act as a building block for various pharmacologically active scaffolds. In the present work, we have reviewed the recent literature pertaining to the synthesis and bioactivity of numerous 8-HQ derivatives as anticancer, antiviral, antimicrobial, antibacterial, antifungal, and anticancer agents. The results obtained from this review highlight the importance of numerous derivatives of 8-HQ as possible chemotherapeutic agents and as possible leads towards the development of new drugs to treat various diseases, including cancer. We hope that data presented in this review could help researchers in the fields of medicinal chemistry and pharmacology in designing new active compounds and in the modification of existing compounds in the search for new drug leads. The development of drugs, either natural or synthetic, is gaining popularity in the fight against diseases, such as cardiovascular disorders, cancer insurgence, and immune dysfunction. There are certain nuclei such as 8-HQ that are important building blocks in the medicinal arena. Therefore, new synthetic methods for bioactive 8-HQ derivatives should be pursued. In this respect, more biological testing, including in vivo studies, should accompany these syntheses. Studies should also involve different pharmacokinetic parameters related to the safety profiles of potent derivatives. In this review, we have shown different synthetic strategies for pharmaceutically important chemicals that incorporate the 8-HQ moiety. These compounds exhibited a wide range of biological activities and could be used as therapeutic agents against different diseases, including cancer. Some of these compounds could be envisioned as leads in the development of drugs.

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
