# Peer review of "Recent Advances in the Synthesis and Biological Activity of 8-Hydroxyquinolines"

_molecules, 2020, doi:10.3390/molecules25184321_

Round 1

Reviewer 1 Report

This review paper deals with the progress in the synthesis and applications of 8-hydroxyquinolines. The syntheses and therapeutic application of hydroxyquinolines has been a topic of considerable interest. The contents in this paper are classified on the basis of type of diseases and it is well-organized. I support publication in Molecules as an excellent review ; however, correction of the following points is needed.

General:

1) The authors should correct “°C (degree centigrade, degree Celsius)”.

2) Each number in R1, R2, … should be written in superscript.

Note:

1) Page 2, Figure 1; Acid-base reactions, Diazonium coupling -> Acid-Base Reactions, Diazonium Coupling

2) Page 2, line 65; With COVID-19 pandemics -> With the COVID-19 pandemics

3) Page 4, Scheme 3; each number in CH2Cl2, K2CO3, and CH3 should be written in subscript.

Author Response

Reviewer 1

General

1) The authors should correct “°C (degree centigrade, degree Celsius)”.

Response:  Corrected.

2) Each number in R1, R2, … should be written in superscript.

Response: Corrections have been made on the revised manuscript.

Note

1) Page 2, Figure 1; Acid-base reactions, Diazonium coupling -> Acid-Base Reactions, Diazonium Coupling

Response: Corrected.

2) Page 2, line 65; With COVID-19 pandemics -> With the COVID-19 pandemics

Response: Corrected.

3) Page 4, Scheme 3; each number in CH2Cl2, K2CO3, and CH3 should be written in subscript.

Response: Corrected.

Reviewer 2 Report

This review concerns the article entitled: Recent Advances in the Synthesis and Biological Activity of 8-Hydroxyquinolines (Manuscript ID: molecules-924415).

This is review type of article with total 67 references covering period 2017 up to present. Apart of short Introduction the main text is divided according to biological activity of the presented compounds: Antiviral, Antibacterial, Antifungal, Anticancer, and Alzheimer. The structure of the compounds together with the synthetic routes are presented in figures and schemes.

The article can be accepted for publication, however, the following remarks should be considered in the revised version:

  1. The Abstract and Conclusion do not contain much information. The abstract should be more concise. On the other hand, the conclusion are too general. It would be better to draw some specific conclusion, which results directly from the described literature survey. In my opinion both Abstract and Conclusion should be reconsidered.
  2. The structures in figures and schemes should be reorganized for convenience of readers. It means the main fragment, 8-hydroxyquinoline, should be drawn always in the same way. For example, pyridynium ring on the left side, phenolic part on the right side. The nitrogen and oxygen atom should be at the bottom. Dimension of structures and letters in figures and schemes should be unified. Instead of divine the pictures in schemes and figures, I suggest all them describes as figures. In the case of reaction, the legend should begin as scheme of … .
  3. Minor remarks. Please, avoid the word “recently” in the text. Scheme 1; style of description above and below arrows should be corrected (mol %, Celsius degree). Page 2, line 70; hyroxquinolines, please correct. Scheme 2; R substituents are not listed, MW shortcut is not described. Scheme 3; subscript should be introduced in molecular formulas. Page 5, line 158; dihydro-2H-[1,3]oxazino[5,6-h]quinolin-2-one should be checked (most probably H and italic). Page 7, line 221; TBAB shortcut should be explained. The passage concerning Figure 2 is very short, it requires some improvement, or if possible Figures 2 and 2 should be joined. Figure 3; the substituent R should be presented in the figure, not only in the text. Page 12, line 325; please, check the compound name 2-amino-4-aryl-4-pyrano[3,2-h]quinolone-3-carbonitril (3,2-h, it should be capital letter and italic). Scheme 13; please, improve the text above and beneath the arrows in the scheme. Scheme 14; please, improve the text above and beneath the arrows in the scheme. Page 15, line 402; please, check compound name 4-aryl-pyrano[3,2-h]quinolones, see also Scheme 16. Page 19, line 529; R2 = cyclopentylethyl, line 531, R1 = cyclopentylmethyl, line 538, R2 = cyclopentylmethyl, please, write the name of substituent instead of structure. Page 20, line 560; please, check the name of compound (8-hydroxyquinolin-5-yl)methyl)phosphonate (lack of bracket). Figure 6; there is no need to use three arrows, just one and above a,b,c.

Author Response

Reviewer 2

The article can be accepted for publication, however, the following remarks should be considered in the revised version:

  1. The Abstract and Conclusion do not contain much information. The abstract should be more concise. On the other hand, the conclusion are too general. It would be better to draw some specific conclusion, which results directly from the described literature survey. In my opinion both Abstract and Conclusion should be reconsidered.

Response: Abstract and conclusions have been revised.

  1. The structures in figures and schemes should be reorganized for convenience of readers. It means the main fragment, 8-hydroxyquinoline, should be drawn always in the same way. For example, pyridynium ring on the left side, phenolic part on the right side. The nitrogen and oxygen atom should be at the bottom. Dimension of structures and letters in figures and schemes should be unified. Instead of divine the pictures in schemes and figures, I suggest all them describes as figures. In the case of reaction, the legend should begin as scheme of … .

  • Response:
  • 1) 8-Hydroxyquinoline drawn with pyridinium ring on the left side and phenolic part on the right side, and The nitrogen and oxygen atoms are at the bottom throughout the review.
  • 2) Dimension of structures and letters in figures and schemes have been unified throughout the manuscript.

  1. Minor remarks. Please, avoid the word “recently” in the text.

  • Response: The word “recently” was removed from the text.

Scheme 1; style of description above and below arrows should be corrected (mol %, Celsius degree).

Response: The style of description above and below arrows has been corrected.

Page 2, line 70; hyroxquinolines, please correct.

Response: Corrected.

Scheme 2; R substituents are not listed, MW shortcut is not described.

  • Response: In Scheme 2, R substituents have been listed, and MW shortcut described as “Microwave (MW)

Scheme 3; subscript should be introduced in molecular formulas.

Response: Done.

Page 5, line 158; dihydro-2H-[1,3]oxazino[5,6-h]quinolin-2-one should be checked (most probably H and italic).

Response: Checked and corrected; dihydro-2H-[1,3]oxazino[5,6-h]quinolin-2-one has been changed to “dihydro-2H-[1,3]oxazino[5,6-h]quinolin-2-one

 Page 7, line 221; TBAB shortcut should be explained.

Response: TBAB shortcut has been explained in the revised manuscript.

The passage concerning Figure 2 is very short, it requires some improvement, or if possible Figures 2 and 2 should be joined.

Response: Passage has been improved and new data have been added.

 Figure 3; the substituent R should be presented in the figure, not only in the text.

Response: Substituent R has been presented in the figure.

Page 12, line 325; please, check the compound name 2-amino-4-aryl-4-pyrano[3,2-h]quinolone-3-carbonitril (3,2-h, it should be capital letter and italic).

Response: We have checked the name; a letter h should be italic but not capitalized.

Scheme 13; please, improve the text above and beneath the arrows in the scheme.

Response: Text has been revised.

Scheme 14; please, improve the text above and beneath the arrows in the scheme.

Response: Text and scheme have been revised.

Page 15, line 402; please, check compound name 4-aryl-pyrano[3,2-h]quinolones, see also Scheme 16.

Response: Page 15 and scheme 16, the name was checked and corrected in the revised manuscript.

Page 19, line 529; R2 = cyclopentylethyl, line 531, R1 = cyclopentylmethyl, line 538, R2 = cyclopentylmethyl, please, write the name of substituent instead of structure.

Response: Done.

Page 20, line 560; please, check the name of compound (8-hydroxyquinolin-5-yl)methyl)phosphonate (lack of bracket).

Response:  Name has been checked and corrected.

 Figure 6; there is no need to use three arrows, just one and above a,b,c.

             Response: Corrected as suggested by the reviewer.

Reviewer 3 Report

Report: Molecules 9254415

  1. Lns 35&36: What exactly is meant by “four and six covalent complex”?
  2. Fig 1 The pyridine ring is usually drawn on the right hand side
  3. Ln 65 ..with the Covid-19 pandemic…
  4. Ln 66   remove “on the Globe”
  5. Ln 71 Please check all spelling of   hydroxyquinolines
  6. Scheme 1   This should rather indicate the 2,6-dichloro   product
  7. Scheme 1 Drawing is too squashed and temp is incorrectly given
  8. For this and the rest of the review: All compounds must be assigned numbers in chronological order throughout and the text should then reflect this as currently it is totally unacceptable and confusing at times.
  9. Ln 76: what “two” derivatives are being referred to? Ln 88 must be rewritten
  10. Scheme 2   What is R?
  11. Ln 114 must be rewritten
  12. Scheme 3 Formulae for DCM and the salt are incorrectly given
  13. Ln 158 rewrite
  14. Ln 160   there is a missing reagent
  15. Ln 170/171 all concentrations should be 1x10-6 and the same for the others ie 1x10…. Nd not just 10
  16. Ln 193 incorrect use of brackets
  17. Scheme Suddenly the authors start using numbers in the schemes. This must be from the very beginning. R1 and R2 must be consistently used
  18. Lns 281-286 Why bare no values given for activities?
  19. Scheme 11 Size of structures different which is unacceptable. Use same drawing programme as he sugars are all over the place.
  20. Scheme 13 for compound 1 the NH3  must be changed to NH2
  21. Lns 351 and 369 rewrite
  22. Scheme 14 What is R?
  23. Scheme 15 What is R?
  24. Scheme 16 Must indicate the Ar ring where the R1-R4 are attached
  25. Ln 448 no structure of compound given?
  26. Ln 464 makes no sense
  27. Fig 15 and some others: Why is no synthesis given as this after all the thrust of the review????
  28. Lns 506-542 Renumbering is required
  29. Schemes 19-22 Renumbering required to be reflected in the text
  30. Scheme 23 Arrows of mechanism must be revised and sizes of molecules are different

Author Response

Reviewer 3

  1. Lns 35&36: What exactly is meant by “four and six covalent complex”?

  • Response: The term “four and six covalent complex” means “four and six coordinative covalent complex”. In coordination chemistry the bond in [M(H2O)n]m+ the bonding between water and the metal cationis described as a coordinate covalent bond. When two 8-hydroxyquinoline molecules form a complex with a metal through N and O is called four coordinative covalent complex. With three molecules of 8-HQ with one metal is called six coordinative covalent complex.

  1. Fig 1 The pyridine ring is usually drawn on the right hand side

Response: For the sake of consistency, and as suggested by reviewer 2, we have drawn the structure with the pyridine ring on the left hand side.

  1. Ln 65 ..with the Covid-19 pandemic…

Response: Corrected.

  1. Ln 66   remove “on the Globe”

Response: Removed.

  1. Ln 71 Please check all spelling of   hydroxyquinolines

Response: Corrected.

  1. Scheme 1   This should rather indicate the 2,6-dichloro   product

Response: It is actually the 5,7-dichloroproduct which is indicated..

  1. Scheme 1 Drawing is too squashed and temp is incorrectly given

Response: Corrected.

  1. For this and the rest of the review: All compounds must be assigned numbers in chronological order throughout and the text should then reflect this as currently it is totally unacceptable and confusing at times.

  • Response: All compounds have been assigned numbers in chronological order throughout the text.

  1. Ln 76: what “two” derivatives are being referred to? Ln 88 must be rewritten

  • Response: The “two” derivatives are (R= i-Pr and i-Bu)” and were added to the text. In addition, Line 88 was rewritten.

  1. Scheme 2   What is R?

            Response: R has been described and included in the scheme.

  1. Ln 114 must be rewritten

Response: Done.

  1. Scheme 3 Formulae for DCM and the salt are incorrectly given

Response: Formula of DCM and the salt have been corrected.

  1. Ln 158 rewrite

Response: Line has been rewritten.

  1. Ln 160   there is a missing reagent

             Response: the missing reagent has been added

  1. Ln 170/171 all concentrations should be 1x10-6 and the same for the others ie 1x10…. Nd not just 10

Response: Corrected.

  1. Ln 193 incorrect use of brackets

Response: Corrected.

  1. Scheme Suddenly the authors start using numbers in the schemes. This must be from the very beginning. R1 and R2 must be consistently used

  • Response: Compounds in Schemes have been numbered and R groups were described.

  1. Lns 281-286 Why bare no values given for activities?

Response: Values for activities have been provided.

  1. Scheme 11 Size of structures different which is unacceptable. Use same drawing programme as he sugars are all over the place.

  • Response: Structures have been properly redrawn.

  1. Scheme 13 for compound 1 the NH must be changed to NH2

            Response: Corrected.

  1. Lns 351 and 369 rewrite

Response: Done.

  1. Scheme 14 What is R?

Response: R has been described.

  1. Scheme 15 What is R?

Response: R has been described.

  1. Scheme 16 Must indicate the Ar ring where the R1-Rare attached

Response: Ar ring was clarified.

.

  1. Ln 448 no structure of compound given?

Response:

  1. Ln 464 makes no sense

  • Response: The structure of 5,7-dibromo-8-hydroxyquinoline was not provided since we think it is simple, and will affect numbers of other compounds in the manuscript.

  1. Fig 15 and some others: Why is no synthesis given as this after all the thrust of the review????

Response: In most cases, synthesis and schemes were given unless the synthesis is a straight forward procedure and does not involve new reagents or conditions, or if the synthesis involves compounds that could be obtained from commercial sources.

  1. Lns 506-542 Renumbering is required

Response: Done.

  1. Schemes 19-22 Renumbering required to be reflected in the text

Response: Done.

  1. Scheme 23 Arrows of mechanism must be revised and sizes of molecules are different

  • Response: Arrows and sizes of molecules have been revised.

Round 2

Reviewer 2 Report

This review concerns the revised version of Manuscript ID: molecules-924415.

Author basically improve the text according to the remarks included in the previous review. In particular, the changes have been made in schemes and figures, the names of compounds as well as some style. However, the suggested major changes Abstract and Conclusion are merely done, therefore, the work can be accepted, but its scientific sound is rather moderate.

Reviewer 3 Report

Please check on the English laguage again. The yellow highlight in the abstarct needs some attention as do other sentences. The science is now acceptable.